# Survival Benefit of Surgical Treatment for Elderly Patients with Intrahepatic Cholangiocarcinoma: A Retrospective Cohort Study in the SEER Database by Propensity Score Matching Analysis

**Kaiyu Chen** [1,2], **Haitao Yu** [1,2], **Jinhuan Yang** [1,2], **Zhiyuan Bo** [1,2], **Chen Jin** [3], **Lijun Wu** [1,2], **Yi Wang** [3,*] **and Gang Chen** [1,2,*]

1   Department of Hepatobiliary Surgery, The First Affiliated Hospital of Wenzhou Medical University, Wenzhou 325015, China
2   Key Laboratory of Diagnosis and Treatment of Severe Hepato-Pancreatic Diseases of Zhejiang Province, The First Affiliated Hospital of Wenzhou Medical University, Wenzhou 325015, China
3   Department of Epidemiology and Biostatistics, School of Public Health and Management, Wenzhou Medical University, Wenzhou 325015, China
*   Correspondence: wang.yi@wmu.edu.cn (Y.W.); chen.gang@wmu.edu.cn (G.C.)

**Abstract:** Despite a rising trend in intrahepatic cholangiocarcinoma (ICC) incidence in the elderly population worldwide, the benefit of surgery for those patients is still controversial. Data from 811 elderly patients diagnosed with non-metastatic ICC were obtained from the US surveillance, epidemiology, and end results (SEER) program database. Propensity score matched (PSM) was conducted for the better balance of baseline. The associations between tumor characteristics and surgery with overall survival (OS) and cancer specific survival (CSS) were estimated using hazard ratios (HR) and 95% confidence intervals (CI). The results showed that ICC patients above 60 years old taking surgery had better OS (hazard ratio [HR], 0.258; 95% CI, 0.205–0.324) and CSS (hazard ratio [HR], 0.239; 95% CI, 0.188–0.303) than patients without surgery. Similar trends in patients above 65 years old, above 70 years old, above 75 years old, and above 80 years old were observed, separately. This benefit was also showed in lymph node-negative (N0) and lymph node-positive (N1) subgroups and N0 patients are more likely to take an advantage from surgery than N1 patients. The different outcomes between surgery and non-surgery suggest that surgical treatment may be recommended for elderly ICC if the tumor is resectable to ensure optimal treatment.

**Keywords:** cholangiocarcinoma; surgery outcomes; survival prognosis; surveillance; epidemiology; end results (SEER) database; older patients

## 1. Introduction

Intrahepatic cholangiocarcinoma (ICC) is a highly malignant carcinoma, accounting for approximately 10% to 20% of the incidence rate of primary liver malignant tumors [1,2]. The incidence rate of ICC has been increasing worldwide during the last decades [3,4]. According to the record of the U.S. surveillance, epidemiology, and end results (SEER) database, the statistics showed that the incidence rate of ICC has increased by more than 120% in the past 40 years [5], which may be attributed to improvements in the diagnosis such as better image techniques and better awareness of this malignant tumor. For patients of ICC, a large proportion are diagnosed between the age of 50 and 70; the average age is approximately 60 years old [6,7]. This age distribution is consistent with the worldwide trend of aging population [8,9], suggesting the need for more attention to the treatment of elderly patients with ICC.

Liver resection is currently the most effective curative therapy for ICC. The postoperative 5-year survival for patients with ICC of all age ranges is from 10% to 40% [10–12]. For

post-surgery relapse and unresectable tumors, systemic therapy such as chemotherapy, radiation therapy, immunotherapy, and targeted therapy also have a wide range of potential applications [12,13]. Elderly patients who need to undergo liver surgery should be selected carefully, considering the balance of the benefit and the harm in elderly patients. On the one hand, some studies have shown that elderly patients could receive a survival benefit from surgery such as liver resection because the risk is low and it is feasible for patients [14–17]. On the other hand, surgery for older patients often poses fatal risks, such as cardiac and malignancy-related complications [18–21]. Especially since the high aggressiveness of ICC, surgical operation needs to follow extensive demolition interventions to obtain much wider surgical margins than HCC and less metastasis. Some studies have shown that surgery on elderly patients should be considered cautiously for poor tumor grades and a higher incidence of complications [22,23]. Age seems to be slightly controversial for liver resection; however, ICC data regarding this topic are scarce. The impact of surgery on the short-term and long-term prognosis of elderly patients with ICC remains to be investigated.

This study aimed to assess the survival benefits of surgical resection in elderly patients with ICC. We conducted a population-based retrospective cohort study to explore the effectiveness of surgical treatment for elderly patients with ICC with overall survival (OS) and cancer-specific survival (CSS). The RECORD reporting checklist was used for this article.

## 2. Materials and Methods

### 2.1. Data Source and Study Cohort

Patients with ICC were selected from the SEER database using SEER*Stat 8.3.8 software (National Cancer Institute, Bethesda, MD, USA); the ID number we used to obtain the access to the SEER database was 12131-Nov2019. The SEER database is a retrospective source of information on cancer incidence and survival provided by the National Center for Health Statistics, covering approximately 35% of the United States population since 1973 [24].

All procedures in this study were performed in accordance with the RECORD guidelines [25]. Patients aged >60 years at diagnosis with primary ICC between 2010 and 2015 were extracted. Patients with ICC were identified by the International Classification of Disease for Oncology 3rd edition, topography code C22.0 (primary liver cancer) with histological code 8160 (cholangiocarcinoma) and topography code C22.1 (intrahepatic bile duct cancer) with histological codes 8140 (adenocarcinoma) and 8160 (cholangiocarcinoma) [26,27]. Only patients with a confirmed positive histology primary ICC were included. Patients without recommendations of surgery were excluded from study. Patients who had complications with other malignant tumors or those without sufficient survival data were excluded. Patients with missing surgery or tumor stage information were also excluded.

The clinical tumor node metastasis (TNM) staging of this study was recoded based on the American Joint Committee on Cancer 8th edition (AJCC 8th edition). Therefore, the collaborative staging codes provided by SEER were used to derive the overall stage and individual T, N, and M classifications according to the criteria. The primary endpoints of this study were OS and CSS. Overall survival was defined as the time duration since the diagnosis of ICC to death regardless of the cause or last follow up. The cancer-specific survival was defined as the time since the diagnosis of ICC to cancer-specific death or last follow up.

### 2.2. Tumor Grade Multiple Imputation

Prior to multiple imputations, tumor grade was dichotomized as G1–2 (well-differentiated or moderately differentiated) versus G3–4 (poorly differentiated or undifferentiated). Missing tumor grade was imputed from age, sex, race, clinical TNM stage, surgery, chemotherapy, and radiotherapy by logistic regression. The imputation was repeated, and the outcome between the missing data and imputed data was examined by comparing the density distribution. Rubin's rule was used to calculate outcomes [28].

### 2.3. Propensity Scores Matching Analysis

Propensity score matching (PSM) was applied to improve the comparability between the surgical and nonsurgical groups. Age at diagnosis, sex, race, grade, TNM stage, chemotherapy, and radiotherapy were selected to calculate the propensity score based on clinical experience or the outcome of prior survival analysis. The propensity score for each case was estimated from the available data using a logistic regression model.

### 2.4. Statistical Analysis

The baseline characteristics of patients in the surgical and nonsurgical groups were compared between those of the unmatched and matched cohorts. Numerical variables are presented as the mean $\pm$ standard deviation (SD) and were examined using the independent sample t-test. Categorical variables are presented as numbers (percentages) and were tested using the Pearson $\chi^2$ test. OS, CCS, and median survival time were calculated using the Kaplan–Meier method. The Cox proportional hazards regression model was used for univariate and multivariate analyses, and the covariables with statistical significance in univariate analysis were selected for the multivariate analysis. $p$-values < 0.05 (two-tailed) were considered statistically significant. R software version 4.0.2 (R Foundation for Statistical Computing, Vienna, Austria) was used for statistical analysis. PSM was performed using the R package "Matching" [29].

## 3. Results

### 3.1. Demographic and Clinicopathologic Characteristics

Of the 5790 patients with primary ICC identified from the SEER database, 1345 patients met the inclusion criteria (Figure S1), of whom 375 (28%) underwent surgery, and 970 (72%) did not. We compared the TNM stage distribution between the two groups and found that it was unbalanced, especially for stage IV (Table S1). Therefore, we distinguished patients with distant metastasis (M1) from those without distant metastasis (M0). Finally, 881 patients with non-distant metastatic ICC were included in the final analysis, of whom 349 patients (40%) underwent surgery and 532 patients (60%) did not. A comparison of demographic and clinical characteristics between the surgery and non-surgery cohorts is presented in Table 1.

**Table 1.** Baseline characteristics of ICC patients above 60 years old before and after propensity score matching.

| Characteristic | Unmatched | | | | Matched | | | |
|---|---|---|---|---|---|---|---|---|
| | ALL (n = 881) | Non-Surgery (n = 532) | Surgery (n = 349) | *p*-Value | ALL (n = 498) | Non-Surgery (n = 249) | Surgery (n = 249) | *p*-Value |
| Age, years | 70.9 ± 7.52 | 71.8 ± 7.96 | 69.5 ± 6.59 | <0.001 | 70.3 (6.91) | 70.4 (7.31) | 70.1 (6.50) | 0.632 |
| Gender | | | | 0.074 | | | | 0.654 |
| Female | 453 (51.4) | 287 (53.9) | 166 (47.6) | | 258 (51.8) | 132 (53.0) | 126 (50.6) | |
| Male | 428 (48.6) | 245 (46.1) | 183 (52.4) | | 240 (48.2) | 117 (47.0) | 123 (49.4) | |
| Ethnicity | | | | 0.839 | | | | 0.912 |
| White | 706 (80.1) | 428 (80.5) | 278 (79.7) | | 396 (79.5) | 197 (79.1) | 199 (79.9) | |
| Non-white | 175 (19.9) | 104 (19.5) | 71 (20.3) | | 102 (20.5) | 52 (20.9) | 50 (20.1) | |
| Grade | | | | <0.001 | | | | 0.927 |
| Well-moderately | 505 (57.3) | 278 (52.3) | 227 (65.0) | | 306 (61.4) | 154 (61.8) | 152 (61.0) | |
| Poorly-undifferentially | 376 (42.7) | 254 (47.7) | 122 (35.0) | | 192 (38.6) | 95 (38.2) | 97 (39.0) | |
| TNM stage | | | | <0.001 | | | | 0.398 |
| IA | 133 (15.1) | 54 (10.2) | 79 (22.6) | | 77 (15.5) | 31 (12.4) | 46 (18.5) | |
| IB | 133 (15.1) | 85 (16.0) | 48 (13.8) | | 70 (14.1) | 37 (14.9) | 33 (13.3) | |
| II | 319 (36.2) | 212 (39.8) | 107 (30.7) | | 182 (36.5) | 97 (39.0) | 85 (34.1) | |
| IIIA | 14 (1.6) | 8 (1.5) | 6 (1.7) | | 11 (2.2) | 6 (2.4) | 5 (2.0) | |
| IIIB | 282 (32.0) | 173 (32.5) | 109 (31.2) | | 158 (31.7) | 78 (31.3) | 80 (32.1) | |
| Chemotherapy | | | | 0.008 | | | | 0.589 |
| No | 483 (54.8) | 272 (51.1) | 211 (60.5) | | 273 (54.8) | 140 (56.2) | 133 (53.4) | |
| Yes | 398 (45.2) | 260 (48.9) | 138 (39.5) | | 225 (45.2) | 109 (43.8) | 116 (46.6) | |
| Radiotherapy | | | | 0.005 | | | | 0.514 |
| No | 727 (82.5) | 423 (79.5) | 304 (87.1) | | 430 (86.3) | 218 (87.6) | 212 (85.1) | |
| Yes | 154 (17.5) | 109 (20.5) | 45 (12.9) | | 68 (13.7) | 31 (12.4) | 37 (14.9) | |

Patients with ICC who underwent surgery were younger than those who did not undergo surgery (69.52 [SD 6.59] vs. 71.77 [SD 7.96] years old, $p$ < 0.001). Moreover, the

surgery patient cohort had a higher percentage of men (52.4% vs. 46.1%, *p* < 0.001), a lower percentage of chemotherapy (39.5% vs. 48.9%, *p* = 0.008), a lower percentage of radiotherapy (12.9% vs. 20.5%, *p* = 0.006), and a lower percentage of G3–4 grade than the non-surgery patient cohort (35.0% vs. 47.7%, *p* < 0.001). The TNM stage distribution in the surgery and non-surgery cohorts is shown in Table S1.

To thoroughly investigate the benefit of surgery for elderly patients with ICC, we further analyzed elderly patients starting from different ages: 669 patients aged >65 years, 454 patients aged >70 years, 272 patients aged above 75 years, and 107 patients aged above 80 years. In the analytical cohort of patients aged 65, 70, 75, and 80 years, the distribution of demographic and clinicopathological characteristics was also unbalanced between the surgery and non-surgery cohorts. To equilibrate these differences in characteristics at the baseline, the propensity score of the surgery and non-surgery case pairs were obtained in each age stratification for further analysis (249 pairs in patients aged >60 years, 194 pairs in patients aged above 65 years, 142 pairs in patients aged above 70 years, 76 pairs in patients aged >75 years) (Tables S2–S4).

### 3.2. Surgery Benefit for Patients with Nonmetastatic ICC

For all patients aged >60 years, the median follow-up OS was 36 months (range, 0.5 to 83 months), which was 37 (range, 0.5 to 83 months) for the surgery cohort and 33 (range, 0.5 to 82 months) for the non-surgery cohort. The median follow-up CSS was 33 months (range, 0.5 to 83 months), which was 35 (range, 0.5 to 83 months) for the surgery cohort and 30 (range, 0.5 to 82 months) for the non-surgery cohort.

In the matched cohort, for patients aged >60 years, there was a benefit for surgery groups compared with the non-surgery group in OS (median survival time, 33 [range, 0.5 to 83] months vs. 8 [range, 0.5 to 82] months; *p* < 0.001) and CSS (median survival time, 39 [range, 0.5 to 83] months vs. 8 [range, 0.5 to 82] months; *p* < 0.001) (Figure 1).

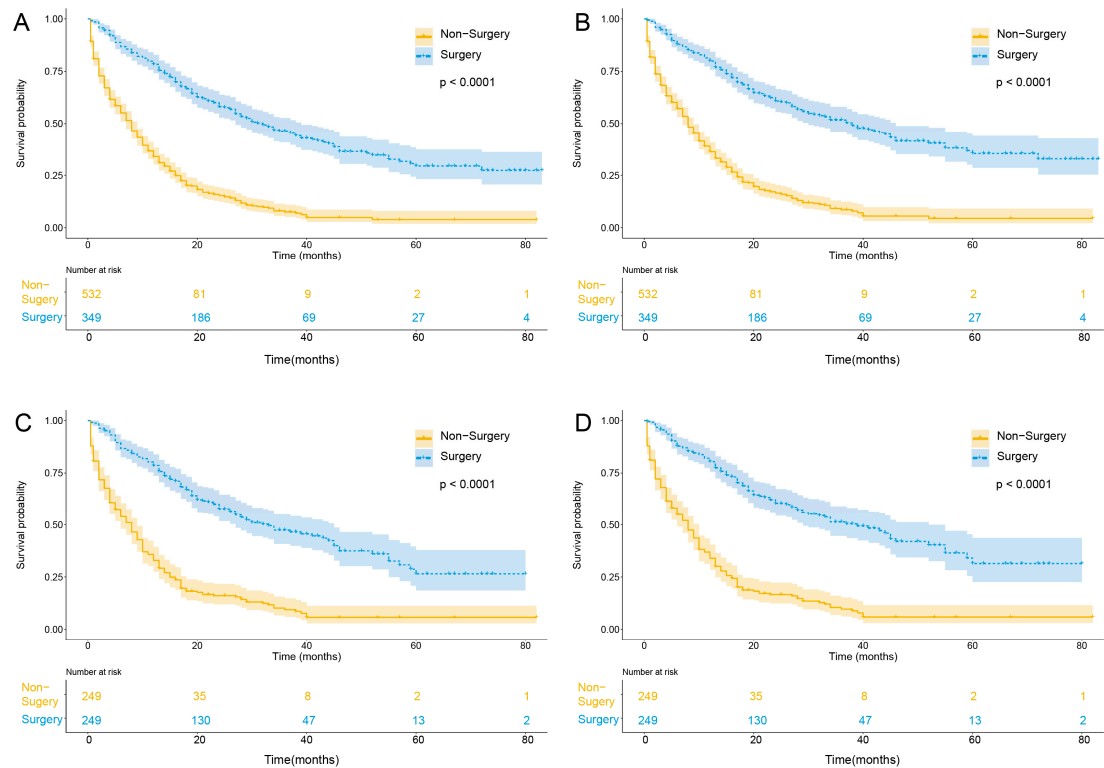

**Figure 1.** Kaplan–Meier curves of unadjusted OS (**A**), unadjusted CSS (**B**), adjusted OS (**C**), and adjusted CSS (**D**) stratified according to surgical treatment in M0 ICC patients above 60 years old. Abbreviations: OS, overall survival; CSS cancer-specific survival; M0, non-distant metastasis.

The univariate analysis of the matched cohort showed that the surgery group had a significantly better OS (hazard ratio [HR], 0.280; 95% confidence interval [CI], 0.224–0.350; $p < 0.001$) and CSS (HR, 0.257; 95% CI, 0.204–0.325; $p < 0.001$) than the non-surgery group. In the data set, other significant clinicopathological parameters for OS and CSS were TNM stage 2–4, presence of chemotherapy, presence of radiotherapy, and poorly undifferentiated grade ($p < 0.05$) (Tables S8–S15, Figures S2 and S3). The multivariate analysis also indicated that surgery was an independent favorable prognostic factor for patients >60 years old in OS (HR, 0.258; 95% CI, 0.205–0.324; $p < 0.001$) and CSS (HR, 0.239; 95% CI, 0.188–0.303; $p < 0.001$); for patients >65 years old in OS (HR, 0.227; 95% CI, 0.172–0.298; $p < 0.001$) and CSS (HR, 0.217; 95% CI, 0.163–0.539; $p < 0.001$); for patients >70 years old in OS (HR, 0.233; 95% CI, 0.171–0.318 $p < 0.001$) and CSS (HR, 0.209; 95% CI, 0.150–0.291; $p < 0.001$); and for patients >75 years old in OS (HR, 0.222; 95% CI, 0.148–0.334; $p < 0.001$) and CSS (HR, 0.457; 95% CI, 0.251–0.832; $p < 0.001$) (Table 2).

The 3-year and 5-year OS and CSS rates also confirmed the superiority of surgical treatment over non-surgical treatment for patients >60 years old (3-year OS, 47.4% vs. 9.4%, $p < 0.01$; 3-year CSS, 51.3% vs. 9.7%, $p < 0.01$; 5-year OS, 26.5% vs. 5.7%, $p < 0.01$; 5-year CSS, 34.4% vs. 5.9%, $p < 0.01$); for patients >65 years old (3-year OS, 42.8% vs. 8.1%, $p < 0.01$; 3-year CSS, 47.7% vs. 8.5%, $p < 0.01$; 5-year OS, 29.0% vs. 3.0%, $p < 0.01$; 5-year CSS, 33.8% vs. 3.2%, $p < 0.01$); for patients >70 years old (3-year OS, 43.3% vs. 5.3%, $p < 0.01$; 3-year CSS, 51.2% vs. 5.7%, $p < 0.01$); and for patients >75 years old (3-year OS, 38.4% vs. 2.2%, $p < 0.01$; 3-year CSS, 40.3% vs. 2.7%, $p < 0.01$) (Table 3). The estimated rate of 5-year OS and CSS for patients >70 and 75 years old was not available because of the few patients who had survived after 5 years.

The short-term surgical outcomes can also be serious in elderly patients, so we estimated 3-month and 6-month survival rates based on the matched patients' cohort. The results showed that patients also benefit from a surgical approach in patients >60 years old (3-month OS, 95.2% vs. 67.7%, $p < 0.001$; 3-month CSS, 95.6% vs. 68.1%, $p < 0.001$; 6-month OS, 86.7% vs. 53.8%, $p < 0.001$; 6-month CSS, 87.9% vs. 54.9%, $p < 0.001$); >65 years old (3-month OS, 93.3% vs. 62.1%, $p < 0.001$; 3-month CSS, 93.8% vs. 62.6%, $p < 0.001$; 6-month OS, 84.5% vs. 48.4%, $p < 0.001$; 6-month CSS, 85.5% vs. 50.3%, $p < 0.001$); >70 years old (3-month OS, 92.3% vs. 59.5%, $p < 0.001$; 3-month CSS, 92.9% vs. 61.3%, $p < 0.001$; 6-month OS, 82.4% vs. 45.2%, $p < 0.001$; 6-month CSS, 84.4% vs. 47.3%, $p < 0.001$); and >75 years old (3-month OS, 92.1% vs. 47.8%, $p < 0.001$; 3-month CSS, 92.1% vs. 51.3%, $p < 0.001$; 6-month OS, 80.3% vs. 35.5%, $p < 0.001$; 6-month CSS, 80.3% vs. 43.7%, $p < 0.001$).

This improvement in prognosis was also found in those aged above 65, 70, and 75 years ($p < 0.001$) (Table S5).

### 3.3. Surgery Benefit for Patients with N0 and N1 Stage ICC

To further compare the prognostic outcomes of different subsets, patients were redivided into lymph node-negative (N0) and lymph node-positive (N1) subgroups. PSM was also applied to balance the baseline. In the N0 subgroup, there was a significantly better OS (HR, 0.242; 95% CI, 0.192–0.307; $p < 0.001$) and CSS (HR, 0.225; 95% CI, 0.116–0.289; $p < 0.001$) in the surgery group above 60 years old; OS (HR, 0.218; 95% CI, 0.165–0.289; $p < 0.001$) and CSS (HR, 0.197; 95% CI, 0.147–0.265; $p < 0.001$) in the surgery group above 65 years old; OS (HR, 0.221; 95% CI, 0.153–0.320; $p < 0.001$) and CSS (HR, 0.214; 95% CI, 0.145–0.315; $p < 0.001$) in the surgery group above 70 years old; and OS (HR, 0.181; 95% CI, 0.109–0.299; $p < 0.001$) and CSS (HR, 0.173; 95% CI, 0.104–0.290; $p < 0.001$) in the surgery group above 75 years old.

Interestingly, patients at the N1 stage and patients above 60 years old even with lymph node metastasis may benefit from surgery for long-term outcomes (OS: HR, 0.494; 95% CI, 0.333–0.733; $p < 0.001$; CSS: HR, 0.482; 95% CI, 0.321–0.724; $p < 0.001$). In other age stratification, the result was similar to patients aged above 60 years, the surgery group had better OS and CSS at N0 and N1 stages than the non-surgery group (Table 4).

**Table 2.** Comparison of survival benefit on OS and CSS for ICC patients in different age stratifications before and after matched.

| | Above 60 Years Old | | Above 65 Years Old | | Above 70 Years Old | | Above 75 Years Old | |
|---|---|---|---|---|---|---|---|---|
| | HR (95% CI) | *p*-Value | HR (95% CI) | *p*-Value | HR (95% CI) | *p*-Value | HR (95% CI) | *p*-Value |
| **OS (surgery vs. non-surgery)** | | | | | | | | |
| Unmatched | 0.237 (0.196–0.286) | <0.001 | 0.211 (0.168–0.264) | <0.001 | 0.184 (0.138–0.244) | <0.001 | 0.196 (0.137–0.280) | <0.001 |
| Matched | 0.258 (0.205–0.324) | <0.001 | 0.227 (0.172–0.298) | <0.001 | 0.233 (0.171–0.318) | <0.001 | 0.222 (0.148–0.334) | <0.001 |
| **CSS (surgery vs. non-surgery)** | | | | | | | | |
| Unmatched | 0.228 (0.187–0.278) | <0.001 | 0.198 (0.156–0.251) | <0.001 | 0.206 (0.153–0.277) | <0.001 | 0.190 (0.129–0.279) | <0.001 |
| Matched | 0.239 (0.188–0.303) | <0.001 | 0.217 (0.163–0.539) | <0.001 | 0.209 (0.150–0.291) | <0.001 | 0.457 (0.251–0.832) | <0.001 |

**Table 3.** Estimated 3-year and 5-year survival rates of surgery and non-surgery groups for M0 stage ICC patients in different age stratification.

| | Above 60 Years Old | | | | Above 65 Years Old | | | | Above 70 Years Old | | | | Above 75 Years Old | | | |
|---|---|---|---|---|---|---|---|---|---|---|---|---|---|---|---|---|
| | 3-Year | | 5-Year | | 3-Year | | 5-Year | | 3-Year | | 5-Year | | 3-Year | | 5-Year | |
| | Estimated Rate (%) | *p*-Value | Estimated Rate (%) | *p*-Value | Estimated Rate (%) | *p*-Value | Estimated Rate (%) | *p*-Value | Estimated Rate (%) | *p*-Value | Estimated Rate (%) | *p*-Value | Estimated Rate (%) | *p*-Value | Estimated Rate (%) | *p*-Value |
| **OS** | | <0.01 | | <0.01 | | <0.01 | | <0.01 | | <0.01 | | | | <0.01 | | NA |
| Non-surgery | 9.4 (6.0–14.8) | | 5.7 (2.9–11.2) | | 8.1 (4.5–14.5) | | 3.0 (0.6–15.1) | | 5.3 (2.1–13.3) | | | | 2.2 (0.3–14.1) | | | |
| Surgery | 47.4 (40.8–55.0) | | 26.5 (18.6–37.9) | | 42.8 (35.6–51.4) | | 29.0 (20.7–40.8) | | 43.3 (35.2–53.2) | | 31.3 (22.7–43.3) | | 38.4 (28.1–52.5) | | 31.9 (21.3–47.9) | |
| **CSS** | | 0.01 | | <0.01 | | <0.01 | | <0.01 | | <0.01 | | | | <0.01 | | NA |
| Non-surgery | 9.7 (6.1–15.3) | | 5.9 (3.0–11.6) | | 8.5 (4.7–15.2) | | 3.2 (0.6–15.9) | | 5.7 (2.3–14.2) | | | | 2.7 (0.4–17.2) | | | |
| Surgery | 51.3 (44.6–59.0) | | 31.4 (22.6–43.6) | | 47.7 (40.3–56.3) | | 33.8 (24.6–46.5) | | 51.2 (42.8–61.3) | | 38.6 (28.7–51.9) | | 43.0 (32.2–57.4) | | 38.7 (27.1–55.2) | |

**Table 4.** Estimated 3-year and 5-year survival rates of surgery and non-surgery groups for N0 and N1 ICC patients in different age stratification.

| | Above 60 Years Old | | | | Above 65 Years Old | | | | Above 70 Years Old | | | | Above 75 Years Old | | | |
|---|---|---|---|---|---|---|---|---|---|---|---|---|---|---|---|---|
| | Univariable | | Multivariable | | Univariable | | Multivariable | | Univariable | | Multivariable | | Univariable | | Multivariable | |
| | HR (95%CI) | *p*-Value | HR (95%CI) | *p*-Value | HR (95%CI) | *p*-Value | HR (95%CI) | *p*-Value | HR (95%CI) | *p*-Value | HR (95%CI) | *p*-Value | HR (95%CI) | *p*-Value | HR (95%CI) | *p*-Value |
| **N0** | | | | | | | | | | | | | | | | |
| OS (Surgery vs. non-surgery) | 0.240 (0.190–0.305) | <0.001 | 0.242 (0.192–0.307) | <0.001 | 0.209 (0.159–0.275) | <0.001 | 0.218 (0.165–0.289) | <0.001 | 0.226 (0.157–0.324) | <0.001 | 0.221 (0.153–0.320) | <0.001 | 0.197 (0.122–0.317) | <0.001 | 0.181 (0.109–0.299) | <0.001 |
| CSS (Surgery vs. non-surgery) | 0.221 (0.173–0.283) | <0.001 | 0.225 (0.116–0.289) | <0.001 | 0.186 (0.139–0.248) | <0.001 | 0.197 (0.147–0.265) | <0.001 | 0.203 (0.139–0.298) | <0.001 | 0.214 (0.145–0.315) | <0.001 | 0.186 (0.114–0.304 | <0.001 | 0.173 (0.104–0.290) | <0.001 |
| **N1** | | | | | | | | | | | | | | | | |
| OS (Surgery vs. non-surgery) | 0.597 (0.405–0.879) | 0.009 | 0.494 (0.333–0.733) | <0.001 | 0.508 (0.321–0.803) | 0.004 | 0.342 (0.211–0.554) | <0.001 | 0.399 (0.239–0.667) | <0.001 | 0.300 (0.177–0.508) | <0.001 | 0.741 (0.393–1.398) | 0.355 | | |
| CSS (Surgery vs. non-surgery) | 0.587 (0.394–0.875) | 0.009 | 0.482 (0.321–0.724) | <0.001 | 0.510 (0.315–0.824) | 0.006 | 0.357 (0.215–0.592) | <0.001 | 0.384 (0.222–0.666) | <0.001 | 0.297 (0.169–0.522) | <0.001 | 0.743 (0.374–1.478) | 0.398 | | |

These outcomes regarding the benefit of surgery are also observed in short-term survival estimation. (Tables S6 and S7). The results showed that N0 stage elderly patients >60 years old receive the survival benefit (3-month OS, 96.4% vs. 64.6%, $p < 0.001$; 3-month CSS, 96.8% vs. 65.0%, $p < 0.001$; 6-month OS, 90.0% vs. 54.1%, $p < 0.001$; 6-month CSS, 91.1% vs. 56.0%, $p < 0.001$). For N1 stage elderly patients >60 years old, the survival benefit could also be observed (3-month OS, 88.4% vs. 63.4%, $p < 0.001$; 3-month CSS, 75.1% vs. 60.9%, $p < 0.001$; 6-month OS, 73.7% vs. 57.5%, $p < 0.001$; 6-month CSS, 75.1% vs. 60.9%, $p < 0.001$). These improvements in prognosis were also found in those aged above 65, 70, and 75 years ($p < 0.001$)

*3.4. Surgery Benefit for Patients above 80 Years Old*

To better study the survival benefit for super elderly patients above 80 years old, this age stratification was further explored. There were 26 pairs of patients included after PSM. The super elderly patients receive a significantly better survival benefit from undergoing surgery (OS, HR, 0.220; 95% CI, 0.111–0.437; $p < 0.001$; CSS, HR, 0.260; 95% CI, 0.127–0.531; $p < 0.001$) (Table 5). Patients above 80 years old were also divided to N0 and N1 subgroups for better comparison. However, the number of patients at N1 stage (five with surgery and nineteen without surgery) is too few to calculate; only the N0 stage patients were included in the study. In the N0 subgroup, there was a significantly better OS (HR, 0.201; 95% CI, 0.089–0.454; $p < 0.001$) and CSS (HR, 0.216; 95% CI, 0.095–0.493; $p < 0.001$) in the surgery group compared with the non-surgery group (Table 5). Since, there are few patients without surgery who survive more than 3 years. The 2-year OS and CSS rates were used to confirm the superiority of surgical treatment over non-surgical treatment for patients >80 years old (2-year OS, 50.4% vs. 5.8%, $p < 0.01$; 2-year CSS, 50.4% vs. 8.3%, $p < 0.01$) (Table S16). The estimated 3-month and 6-month survival rates showed that super elderly patients benefit from a surgical approach in the M0 stage (3-month OS, 92.3% vs. 53.9%, $p < 0.001$; 3-month CSS, 92.3% vs. 56.4%, $p < 0.001$; 6-month OS, 84.6% vs. 26.9%, $p < 0.001$; 6-month CSS, 84.6% vs. 38.7%, $p < 0.001$) and the N0 stage (3-month OS, 95.5% vs. 58.0%, $p < 0.001$; 3-month CSS, 95.5% vs. 58.0%, $p < 0.001$; 6-month OS, 86.4% vs. 24.1%, $p < 0.001$; 6-month CSS, 86.4% vs. 29.0%, $p < 0.001$) (Table S17).

**Table 5.** Comparison of survival benefit of OS and CSS for ICC patients above 80 years old.

|  | M0 | | N0 | |
|---|---|---|---|---|
|  | **HR (95% CI)** | ***p*-Value** | **HR (95% CI)** | ***p*-Value** |
| OS (surgery vs. non-surgery) |  |  |  |  |
| Unmatched | 0.309 (0.182–0.525) | <0.001 | 0.261 (0.139–0.488) | <0.001 |
| Matched | 0.220 (0.111–0.437) | <0.001 | 0.201 (0.089–0.454) | <0.001 |
| CSS (surgery vs. non-surgery) |  |  |  |  |
| Unmatched | 0.320 (0.182–0.563) | <0.001 | 0.278 (0.144–0.536) | <0.001 |
| Matched | 0.260 (0.127–0.531) | <0.001 | 0.216 (0.095–0.493) | <0.001 |

## 4. Discussion

Surgical treatment such as liver resection remains the first-line therapy for ICC [11,30], although only 20–30% of patients are resectable when they are diagnosed [31]. Although the number of elderly patients diagnosed with ICC is increasing [32,33], only a few corresponding studies have focused on them. Therefore, it is necessary to evaluate the surgical improvement for elderly patients with ICC and explore predictors of the outcomes. The findings can offer a new guidance in clinical practice for elderly patients.

Liver resections have become increasingly common [34,35]; a report from Japan showed that patients above 70 years old have made up 50% of those undergoing liver resection [15]. A study by Vitale et al. showed that for patients with ICC, although the perioperative complications may increase for elderly patients compared with younger

patients, the long-term OS is comparable [22]. It was also shown by Bartsch et al. that age has no influence on the feasibility of liver surgery in elderly patients [32]. However, these studies only compared the survival benefit between older patients and younger patients and ignored this benefit for elderly patients themselves. One study calculated the incidence rate of complications after hepatectomy for HCC, which is about 10–40%; morbidity rates of elderly patients are not different from those in younger patients [15], demonstrating the feasibility of liver resection on elderly patients.

Our study included 881 elderly patients with ICC; the TNM stage was transformed according to the *AJCC 8th edition* staging system for being better adjusted to the new standards. As a result, our study demonstrated that elderly patients could receive a significantly better long-term OS and CSS survival from surgical resection; even the super elderly patients above 80 years old could benefit from surgery. In this study, we focused on M0 patients, because liver resection is not available when the distant disease occurs. In elderly patients without distant disease, we also explored the survival benefit on N0 and N1 subgroups. The result of this study showed that surgical resection was also a significant prognostic improvement factor in these two subgroups of N0 and N1 stage, regardless of the age group.

One study showed that patients with solitary tumors in the N1 stage were more likely to survive if they underwent surgery instead of not [36], which is consistent with our study in elderly patients. According to a study based on the SEER database, the positive lymph node was observed in 25.2% of cases [26]. The necessity of lymphadenectomy for ICC patients is not clear. Lymphadenectomy did not show a positive impact on OS in the result of a meta-analysis [37]. According to the result of our study, the N1 stage patients reap a significant survival benefit from liver resection whether the lymphadenectomy was taken.

It is worth noting that chemotherapy was also an important factor for improving the prognosis of most subgroups. It is widely accepted that adjuvant therapies are beneficial for survival; therefore, no further analysis was performed for them to investigate the prognosis impact. It is noticed that the targeted therapy and immunotherapy were not used widely before 2015, and the chemotherapy such as gemcitabine or oxaliplatin is mostly used as a postoperative adjuvant chemotherapy. Several studies have shown that N1 stage ICC may have better survival from chemotherapy and radiotherapy [38,39]. With the advent of new systemic and locoregional treatment options, patients may benefit more from perioperative therapy or multiple approaches rather than surgery alone [30,31,40,41]. Thus, the impact of tumor grade and chemotherapy should also be considered for further in-depth studies.

Interestingly, our study found that poorly-differentiated and undifferentiated tumors in patients aged above 60 and 65 years conferred better OS and CSS. The explanation could be that the progression of G3–4 tumors leads to the earlier appearance of symptoms; thus, the liver resection was taken at an early stage.

For the various comorbidities after liver resection in elderly patients, we also estimated the short-term survival benefit on OS and CSS after 3 months and 6 months. The result showed that although the patients may suffer from postoperative complications, the elderly patients with ICC also benefit from surgery.

A previous study reported that even ICC patients at AJCC stage IV may still benefit from surgery [42], although ICC is considered unresectable in intrahepatic or distant metastases [6]. In our study, the M1 stage patients were eliminated because of the difficulty to balance the TNM stage mentioned before and the inconclusive disagreement on surgery.

Retrospective and meta-analytic studies mentioned above have suggested potential bias, such as the clinical experience of clinicians and heterogeneity of the patient cohorts. To better balance the potential bias comparison of the selected clinical and pathological characteristics in the baseline, PSM was employed to construct a matched cohort. The PSM method could better analyze the prognostic survival results and clearly demonstrate the true efficacy of surgery for elderly patients.

This study evaluated the benefit of a surgical approach for oncologic disease in potentially fragile patients (for age). The outcomes demonstrated that the surgical group had a

significantly better prognosis in OS and CSS than the non-surgery group. According to the unmatched and matched groups, the subgroup analysis of N0 and N1 showed that surgery should be considered for elderly patients with or without lymph metastasis.

This study still has some limitations. First, biases were inevitable due to the nature of the retrospective study such as selective bias, although PSM was applied to balance the baseline. Second, the SEER database does not include details on surgery, such as the time duration of surgery and postoperative complications. Therefore, the short-term outcomes, which can also be severe in elderly patients, should be carefully evaluated by these variables. Instead, we analyzed the 3-month and 6-month survival rates as compensation for short-term outcomes. Third, the SEER database lacks specific data of preoperative or postoperative adjuvant therapies, such as chemotherapy and radiotherapy. Additionally, individual-level data regarding genomics, socioeconomic-specific factors, or immuno-nutritional status are not available, which may lead to increased surgical risk for elderly patients.

## 5. Conclusions

In conclusion, our study suggests an OS and CSS benefit for surgical treatment in M0 stage elderly patients compared with non-surgical treatment. In addition, it is worth noting that the survival advantage of liver resection also exists in N0 and N1 stage elderly patients. Here, surgical resection should be considered as a choice if patients are eligible for surgery. For patients who lose the opportunity for local destruction and resection, chemotherapy is recommended.

**Supplementary Materials:** The following supporting information can be downloaded at: https://www.mdpi.com/article/10.3390/curroncol30030201/s1, Table S1: The proportion of TNM stages for TNM I-IV, unmatched TNM -IIIB and matched TNM I-IIIB for ICC patients; Table S2: Characteristics of ICC patients above 65 years old before and after propensity score matching; Table S3: Characteristics of ICC patients above 70 years old before and after propensity score matching: Table S4: Characteristics of ICC patients above 75 years old before and after propensity score matching; Table S5: Estimated 3-month and 6-month survival rates of surgery and non-surgery groups for M0 stage ICC patients in different age stratification; Table S6: Estimated 3-month and 6-month survival rates of surgery and non-surgery groups for N0 stage ICC patients in different age stratification; Table S7: Estimated 3-month and 6-month survival rates of surgery and non-surgery groups for N1 stage ICC patients in different age stratification; Table S8: Univariate and multivariate cox results table for above 60 years old patients of ICC for overall survival; Table S9: Univariate and multivariate cox results table for above 65 years old patients of ICC for overall survival; Table S10: Univariate and multivariate cox results table for above 70 years old patients of ICC for overall survival; Table S11: Univariate and multivariate cox results table for above 75 years old patients of ICC for overall survival; Table S12: Univariate and multivariate cox results table for above 60 years old patients of ICC for cancer-specific survival; Table S13: Univariate and multivariate cox results table for above 65 years old patients of ICC for cancer-specific survival; Table S14: Univariate and multivariate cox results table for above 70 years old patients of ICC for cancer-specific survival; Table S15: Univariate and multivariate cox results table for above 75 years old patients of ICC for cancer-specific survival; Table S16: Estimated 2-year survival rates of surgery and non-surgery groups for ICC patients above 80 years old; Table S17: Estimated 3-month and 6-month survival rates of surgery and non-surgery groups for ICC patients above 80 years old; Figure S1: Flowchart of Patient Selection; Figure S2: Forest plot and summary statistics of the surgery influence on intrahepatic cholangiocarcinoma incidence for OS in ICC; Figure S3: Forest plot and summary statistics of the surgery influence on intrahepatic cholangiocarcinoma incidence for CSS in ICC.

**Author Contributions:** Conception and design: K.C., Y.W. and G.C.; collection and assembly of data: K.C. and Z.B.; data analysis and interpretation: K.C., H.Y., J.Y., C.J. and L.W.; manuscript writing: all authors; final approval of manuscript: all authors; accountable for all aspects of the work: all authors. All authors have read and agreed to the published version of the manuscript.

**Funding:** This research was funded by National Natural Science Foundation of China, grant number 81772628, 81703310, 82072685 and the Research Foundation of National Health Commission of China-Major Medical and Health Technology Project for Zhejiang Province grant number WKJ-ZJ-1706.

**Institutional Review Board Statement:** We received permission from the National Cancer Institute, US to access the research data file in the SEER program (reference number 12131-Nov2019). Ethics approval was not applicable because SEER data is publicly available and anonymized database, patients have been previously de-identified.

**Informed Consent Statement:** As the data are publicly available, no informed consent is required.

**Data Availability Statement:** The datasets analyzed in the current study are available in the SEER repository (https://seer.cancer.gov/ accessed on 12 November 2019) and are publicly accessible.

**Conflicts of Interest:** The authors declare no conflict of interest.

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
