# Peer review of "Survival Benefit of Surgical Treatment for Elderly Patients with Intrahepatic Cholangiocarcinoma: A Retrospective Cohort Study in the SEER Database by Propensity Score Matching Analysis"

_curroncol, doi:10.3390/curroncol30030201_

Round 1

Reviewer 1 Report

Kaiyu Chen et al. present in this manuscript their analysis of Data collected in the SEER (US Surveillance, Epidemiology, and End Results) database of the survival rate of patients with intrahepatic cholangiocarcinoma. They focus on elderly patients (beyond the age of 60) and compared operated versus non-operated patients. 

Their result is that operated elderly patients, even when they have already metastases in local lymph notes, have a significant better survival rate than non-operated patients. They exclude patients with distant metastases because those patients are excluded from operational therapy. 

There is certainly a tendency that patients the are generally fitter are operated with preference. For this reason the authors stratify the patient groups by a propensity score: a score that summarises several factors, like age, race, tumor grade, other therapies obtained to make the operated and non-operated group as comparable as possible. 

The manuscript presents novel findings in the sense that the study focusses on patients older than 60. There are not many studies focussing on this group even though most of the patients belong to it. The content is significant because the results transmit a clear message that it is beneficial for the patient's survival to operate the primary tumor. The article is well written and easy to understand. The presented data support the author's conclusions. So, everything appears to be coherent. The findings are of interest to the reader because the manuscript transmits a recommendation for the treatment of the hepatic cancer. 

Even if some bias concerning overall fitness can not be extracted from the available data the authors try to compare only homogenised groups. The authors admit this in their limitation section. 

In summary, I think the article merits be published as it is. 

Author Response

Many thanks to you for encouraging our work. We are excited about the positive feedback on our manuscript. We will continue the work on the treatment of the hepatic carcinomas. Thanks again for your continued support of this work.

Reviewer 2 Report

The authors described "Survival Benefit of Surgical Treatment for Elderly Patients with Intrahepatic Cholangiocarcinoma: A retrospective cohort study in the SEER database by propensity score matching analysis". This topic should be informative and attractive for potential readers. However, as there are some typos in the manuscript (e.g. line 19 surgy), English editing should be done. Also, how about super elderly patients aged >80 years? Please add it.

Author Response

Thank you very much for the effort and time on our manuscript. Please see the attachment.

Reviewer 3 Report

The manuscript "Survival Benefit of Surgical Treatment for Elderly Patients with Intrahepatic Cholangiocarcinoma: A retrospective cohort study in the SEER database by propensity score matching analysis" from my point of view is written exhaustively in all its sections: introduction, materials and methods, Results and discussion. The references are correct. The results described, obtained on a significant number of patients, provide a good contribution to the scientific community.

Author Response

Thank you for spending precious time on our manuscript. We are encouraged by your positive comments. We will continue the research  and make more progress to contribute to the scientific community.